# The Role of Bacterial Polyhydroalkanoate (PHA) in a Sustainable Future: A Review on the Biological Diversity

**DOI:** 10.3390/ijerph20042959

**Published:** 2023-02-08

**Authors:** Diogo Vicente, Diogo Neves Proença, Paula V. Morais

**Affiliations:** Department of Life Sciences, Centre for Mechanical Engineering, Materials and Processes, University of Coimbra, 3000-456 Coimbra, Portugal

**Keywords:** polyhydroxyalkanoate, bacteria, sustainability, feedstock, metabolism, applications

## Abstract

Environmental challenges related to the mismanagement of plastic waste became even more evident during the COVID-19 pandemic. The need for new solutions regarding the use of plastics came to the forefront again. Polyhydroxyalkanoates (PHA) have demonstrated their ability to replace conventional plastics, especially in packaging. Its biodegradability and biocompatibility makes this material a sustainable solution. The cost of PHA production and some weak physical properties compared to synthetic polymers remain as the main barriers to its implementation in the industry. The scientific community has been trying to solve these disadvantages associated with PHA. This review seeks to frame the role of PHA and bioplastics as substitutes for conventional plastics for a more sustainable future. It is focused on the bacterial production of PHA, highlighting the current limitations of the production process and, consequently, its implementation in the industry, as well as reviewing the alternatives to turn the production of bioplastics into a sustainable and circular economy.

## 1. Brief Introduction to Plastics

Plastics are synthetic polymers and, due to their versatility, this material can be used for several applications [1]. Since the discovery of the first plastic material in 1907 by Bakelite [2], the plastic industry has grown to the point that the material has become one of the most used materials of the 21st century. Several features of plastic such as softness, heat seal ability, good strength-to-weight ratio, and transparency allowed it to substitute different materials, for instance, glass, wood, metals, and others, in diverse functionalities [1,3]. Plastics have other advantages, for example, the possibility of changing their properties by chemical manipulation [4]. Some of the most used conventional plastics are polyethylene (or polyethene, PE), polypropylene (PP), polyvinylchloride (PVC), polystyrene (PS) and poly ethyl terephthalate (PET). These can give rise to a lot of different products used in the most diverse everyday applications [1]. Plastics are mainly used in packaging but are also used in various industries such as building and construction, electronic, textiles, transportation, medicine, agriculture, and others [5].

Nowadays, there is a lot of concern regarding environmental issues with plastics and the environmental pollution generated by their waste being a major focus of attention. The plastics produced today are mostly non-biodegradable, which leads to their accumulation in the environment at the end of their life cycle. Mismanagement of plastic waste has worsened the environmental pollution situation in recent years [6]. Usually, after completing their purpose, plastics have four main paths they can follow: recycling (7%), incineration (8%), and landfill or abandonment in the environment (85%) [7,8]. None of these destinations proves to be ideal. The less used process is recycling since it is a slow and difficult process. Some of the additive substances used in plastic recycling make the process even more complex and challenging [9,10]. Incineration allows some of the energy contained in the materials to be recovered but produces toxic gases. In landfills, the energy in the materials is lost and there are risks of soil and water contamination [9]. Single-use plastics are mainly used in packaging, for example, cups, plates, bottles, bags, containers, and others. These are usually discarded after use and are the main source of plastic pollution [5]. Several studies have reported the negative effect of plastics on the environment and the situation is expected to continue worsening. Moreover, conventional plastics are non-biodegradable, they can be fragmented into smaller portions, by thermal and mechanical processes, such as microplastics or nanoplastics [11].

Micro- and nanoplastics are harmful to the environment and living beings [12]. Many of these microplastics can absorb environmental contaminants which are harmful when enter living organisms [13]. Recent studies have shown that several marine animals from different habitats have microplastics present in their system, and these can be ingested directly or indirectly [14]. A study exploring the biological effects of microplastics on vertebrate species showed that oral delivery in the fish *N. guentheri* shortens lifespan and accelerates the development of age-related biomarkers [15]. In the case of humans, exposure can follow different pathways such as dermatological contact, inhalation, or ingestion. Microplastics were found in different biological end points in humans with different toxic behaviour. Additionally, studies have shown that humans ingest large amounts of contaminant-laden microplastics which in contact with digestive enzymes, pH, and surfactants in the human digestive system significantly increase the biological accessibility of pollutants and pose a substantial risk to human health [16].

New alternatives to petroleum-based plastics have been explored, especially bioplastics that are biodegradable [10]. Bioplastics are generally defined as plastics produced from renewable raw materials, such as starch- or cellulose-based substrates, or by several polymers produced by microorganisms [17]. Polyhydroxyalkanoates (PHA) are polyesters naturally produced and accumulated as granules inside bacteria cells (for example) and their physicochemical and thermal properties depend on each producer organism and cultivation conditions. Poly-3-hydroxybutyrate (PHB) is one of the most common types of PHA synthesized by several bacteria [18]. The biodegradability and biocompatibility make these bio-based plastic products the target to replace conventional plastics in an eco-friendly and sustainable perspective [19]. With this objective, it is important to assess the diversity of bacteria as producers of PHA, their genetic organization and resulting products, and the benefits and limitations of PHA production that will be addressed in this review.

In this review, the contribution of biopolymers as substitutes for synthetic polymers for a more sustainable future is explored. It focuses mainly on one class of biopolymers, the polyhydroxyalkanoates (PHA) and their bacterial production. This is a topic that has been increasingly explored by the scientific community driven by an urgent need for change, and this article seeks to summarize some key aspects of this area.

## 2. COVID-19 Pandemic Increased the Demand for Plastics

Despite the negative consequences of plastic for environmental pollution, in medical care, plastics with single-use applications are required and sometimes mandatory, e.g., reducing the transmission of bloodborne infections such as human immunodeficiency virus (HIV) and hepatitis B [9]. The need to use plastics was more evident because of the COVID-19 pandemic [20]. In late 2019, the first case of COVID-19 was reported. Due to the ease and speed of contamination, COVID-19 had become a global pandemic within months [21]. To protect citizens, several countries have taken measures to combat contagions, such as partial and total shutdowns, mandatory facemask use, and other restrictions. These measures have proved effective in reducing the number of infections, and some environmental improvements have also been documented, notably air and water quality improvements, pollution reduction, and ecological restoration [22]. Although, the environmental challenges related to the mismanagement of plastic product waste have worsened [22,23,24].

The demand for plastics increased sharply during COVID-19. Plastics are used in the production of safety equipment essential for the protection not only of citizens (face masks, gloves, etc.) but also for the manufacture of medical protective suits, aprons, gowns, face shields, surgical masks, and others [20]. Single-use materials have thus increased plastic waste [20,24]. Citizens staying at home have increased online shopping, such as for medicine, food, and groceries [23]. This increased the demand for packaging, which was already considered the main source of plastic pollution before COVID-19 [5]. Mismanagement of infected medical waste promotes an increased risk to human health [24,25]. A recent study shows that about 1.56 billion face masks are likely to enter the ocean in 2020 [26]. Some measures to mitigate the impact of plastic waste, in particular surgical masks, are under consideration. Recently, Li et al. (2022) proposed the use of surgical masks as feedstock for fuels and energy through pyrolysis [27]. This would allow the waste from surgical masks to be valorised, with some benefits and advantages for the environment. However, there are some environmental challenges associated with this process, and further research in this area and into similar solutions is required. Studies have also shown that concern about the environmental pollution generated by plastics increased during the pandemic [28]. Gareiou et al. (2022) stated that citizens in high- and middle-income economies show increased concern about the use of plastic, its end-of-life scenario, and the potential for its replacement [29]. The COVID-19 pandemic has highlighted the associated environmental challenges and the need to find solutions for the use of a material that is necessary for society but so harmful to the environment [29].

One of the most promising and most explored solutions by the scientific community today is the replacement of synthetic plastics with biopolymers. The similar properties between the two are an important factor for this substitution, but some factors are hindering the imposition of these materials in the industry. The next chapters will discuss biopolymers as synthetic plastic substitutes, namely PHA, their producers, advantages and disadvantages and their sustainability.

## 3. Biopolymers as Substitutes for Petroleum-Based Plastics

Biopolymers are natural products that living organisms can produce as a reserve of carbon and nitrogen under specific environmental conditions [30,31]. Biopolymers come from a variety of sources such as concentrate from biomass (polysaccharides, proteins, etc.), chemical synthesis from bio-based monomers (polylactic acid, a bio-polyester), and produced by microorganism activity and through chemical synthesis from both bio-derived monomers and petroleum-based monomers [32]. Bacteria can produce a diverse number of polymers that differ in their block structure and the linkage of their monomers. Some examples of these polymers are polysaccharides, polyesters, polyphosphates, polynucleotides, and polyamides [30,33]. These polymers have important cellular roles such as structural functions and storage of energy and genetic information [33]. Due to their similar properties to some synthetic materials, biopolymers can replace these materials in diverse applications, allowing the reduction of environmental impacts. These materials have interesting properties, two of the most important are their biodegradability and biocompatibility [31,34]. As defined by Raza et al. (2019), biocompatibility is the capacity of materials to execute their applications safely and harmlessly in a host, which makes biopolymers excellent materials for medical use [34]. Nowadays, biopolymers are already used in medicine, the food industry, and packaging [30].

Polymers that have bio-origin can be used in the manufacture of plastics, forming bioplastics. Examples of biopolymers that can produce bioplastics are cellulose, starch, chitosan, chitin, lignin, and microbial polyesters such as polylactic acid (PLA) and PHA [35]. There is a great diversity of bioplastics, differing in their monomer composition, structure size, and physicochemical properties depending on the producer and conditions of production [31]. Bioplastics can be divided into three groups: bio-based and biodegradable, bio-based and non-biodegradable, and petroleum-based and biodegradable, while petroleum-based and non-biodegradable are defined as conventional plastic [1].

The increase in the price of oil, the accumulation of plastic waste in landfills and oceans, the Kyoto protocol, and the news that China would no longer accept plastic waste for recycling were the most important contributing factors to increasing the use of bioplastics, [17,36] and the most recent one, COVID-19 [6,20,24,29]. Bio-based plastics are important to avoid dependence on oil sources. Biodegradable bioplastics will be a major substitute for conventional plastics used in packaging, thus anticipating mainly waste management issues. Non-biodegradable bioplastics will also be relevant in uses that require them to be resistant and long-lasting. It is believed that the use of bioplastics contributes to a better implementation of a circular economy and sustainability. For a process to be considered sustainable, some conditions need to be fulfilled. For example, the continuous use of resources cannot lead to a depletion of the source, and the residues generated by the process also cannot be accumulated in the environment [17]. The use of bioplastics is sustainable insofar as it uses renewable biomass in the production of materials, thus preserving natural resources, namely petroleum sources. Waste can also be used as a resource to produce these materials, further promoting the circulation of matter (Figure 1).

## 4. Polyhydroxyalkanoates (PHA) and Bacterial Producers

PHA are polyesters produced by many Gram-positive and Gram-negative bacteria [33]. The first report of PHA production was in 1926 and was detected in *Bacillus megaterium* [37]. These natural polymers are accumulated in the form of water-insoluble granules of 0.2–0.5 µm diameter inside the cells [35] and serve as a storage material for carbon and energy [33]. Stressors such as fluctuations in temperature, osmolarity, pH, elevated pressure, or the presence of microbial growth inhibitors were reported to affect PHA production [38]. Most known PHA bacterial producers show better polymer production under nutrient limitation (e.g., nitrogen, phosphorus, oxygen, and magnesium) and excess carbon source. Still, some bacterial groups do not need nutrient limitations to produce PHA [10]. Bacteria such as *Alcaligenes lactus*, a mutant strain of *Azotobacter vinelandii*, and recombinant *Escherichia coli* can produce and store PHAs during the growth phase [10,39]. To date, about 92 bacterial genera can produce PHA under both anaerobic and aerobic conditions and more than 150 monomers of PHA are known. Studies show that PHA constitutes a biological advantage for producers when exposed to freezing [40] and other stress conditions such as oxidative and osmotic pressure [41]. *Ralstonia eutropha* H16 is considered the model organism for the study of PHA production [41]. *Pseudomonas* species are also widely studied for PHA production due to their versatility and ability to produce polymers from various carbon sources [42]. A list of various PHA-producing genera, as well as the substrate used for fermentation and the type of PHA produced, are given in Table 1.

PHAs are increasingly arousing the interest of the scientific community and industry. This offers the possibility to replace synthetic plastics, reaching the so-desired circular economy. The aim is to find new forms of this polymer that can fulfil certain properties and functions [108]. The PHA polymers are composed of (R)-hydroxyalkanoic acid (HA) monomers. These monomers can vary in the alkyl side chain (R), which differs in the number of carbons, and this is responsible for the wide variety of PHAs [3,109]. The molecular mass of PHA can vary between 50 kDa and 100 kDa depending on the PHA producer [4]. The general structure of PHAs is represented in Figure 2.

Depending on their chain length, PHAs can be divided into three main groups: short-chain length (scl), medium-chain length (mcl), and long-chain length (lcl). Depending on their monomeric constitution, PHAs can be homopolymers, i.e., constituted by equal monomers, or heteropolymers, constituted by different monomers [10]. Scl-PHAs have 3–5 carbon atoms in their structure. PHB, the most well-studied PHA type, belongs to this group, and it is a homopolymer. Many bacteria such as *Bacillus megaterium, Burkholderia cepacia, Cupriavidus necator*, and others can produce scl-PHAs [18]. Mcl-PHAs are polymers with 6–14 carbons. Some examples of these polymers are poly(3-hydroxyhexanoate) P(3HHx) and poly(3-hydroxyoctanoate) P(3HO) [10]. Some bacteria such as *Pseudomonas aeruginosa*, *P. oleovorans*, and *P. corrugata* can produce mcl-PHAs [18]. Lcl-PHAs have more than 14 carbon atoms and are rarely produced by microorganisms [35]. The physical properties of polymers are distinct in these groups, with scl-PHAs being known to be brittle, while mcl-PHAs have elastomeric properties [4,110].

PHB is the most studied polymer belonging to the PHA class due to its properties similar to the synthetic polymer polypropylene (PP). The composition of the carbon sources used by the microbial producers affects the properties of PHB [111]. However, when compared to synthetic polymers, PHB has disadvantages, namely in its physical properties. The structure of PHA can be altered by physical, chemical, or biological methods to improve its properties. Currently, several PHB copolymers (polymers constituted by 3-hydroxybutyrate and other monomers) are known, which present diversity in their structures and properties, thus being able to be used in a wide range of applications. Examples of PHB copolymers, their properties, and applications are reviewed by Raza et al. (2019) [34]. One of the most used examples of copolymers is poly(3-hydroxybutyrate-co-3-hydroxyvalerate) (P(3HB-co-3HV)), which can be produced by several bacteria [112]. This copolymer has greater flexibility and other characteristics different from PHB. Terpolymers have more than one secondary monomer in their constitution and are also currently being explored as an alternative to copolymers. Examples of these polymers produced by bacteria are P(3HB-co-3HV-co-3HHx) and P(3HB-co-3HV-co-4HB). Recently, a genetically engineered *Ralstonia* strain was able to produce P(3HB-co-3HV-co-HHx) by using tung oil, obtained from the nut seed of the tung tree, as a substrate [113]. The production of copolymers and terpolymers allows for a multitude of polymers with different characteristics, which can be adjusted to achieve the desired properties. As research progresses, new forms of polymers and new types of bonds are being discovered, allowing the spectrum of properties and applications achieved by these polymers to increase. This versatility allows PHA polymers to become an increasingly effective alternative to synthetic polymers in various applications.

## 5. Feedstocks for PHA Production

PHA can be produced from a diversity of substrates. In the industry, the majority of PHAs are produced from sucrose, sugar corn, and vegetable oils [114]. The use of these substrates has two main disadvantages: the competition for substrates with the food industry [114] and the cost of production [115]. Despite its numerous advantages, PHA production is not well established in the industry due to its high production and recovery costs [10,31]. Currently, researchers are trying to use cheaper substrates for PHA production, such as waste feedstocks (WF), lignocellulosic feedstocks (LF), dairy industry waste, oil industry waste, municipal wastes, biodiesel industrial waste, and waste syngas [116]. Some of the recent developments of feedstocks, strains, and associated process developments for PHA production are reviewed in Li and Wilkins (2020). The use of LF for PHA production involves pre-treatment and hydrolysis processes, which allow more availability of carbon sources for microbial fermentation and PHA production [117]. Currently, this is still a process that needs further investigation due to low production efficiencies. Nevertheless, Argiz et al. (2021) studied the use of fish oil waste as a carbon source for PHA production in a two-stage process (culture selection and accumulation of intracellular compounds) taking advantage of the use of mixed microbial cultures (MMC). The substrate is hydrolysed in two phases without the need for prior treatment by bacteria belonging to MMC, releasing soluble free fatty acids (FFA). Subsequently, soluble FFA can enter cells, functioning as a substrate for various metabolic pathways [118]. Volatile fatty acids (VFA) have been used as a carbon source to produce PHA. Different bacterial strains were reported to reach PHA accumulations up to 80% using this carbon source and, therefore, this application is considered an alternative for the reduction of fermentation costs. Recently, Vu et al. (2022) showed the effect of different concentrations of VFAs as a sole carbon source for the biosynthesis of PHAs using *Cupriavidus necator*, formerly known as *Ralstonia eutropha* [119]. Crude glycerol has also been explored for PHA production. This is a by-product produced by the biodiesel industry. Crude glycerol is one of the few waste sources that can be used directly as the substrate for PHA production, therefore, it is considered a promising alternative substrate in reducing production cost [106]. Wen et al. (2020) investigated PHA production dynamics with the different changing directions of the crude glycerol gradient. Reverse glycerol gradient was demonstrated to be more effective in accumulation batch assays using MMC with origins in wastewater treatment tanks [120]. Future studies will be important for optimizing the WF and LF valorisation processes in the PHA production process to make it increasingly sustainable and implemented in the industry.

## 6. Metabolic Pathways Involved in PHA Synthesis

Diverse genomic and metabolic studies allow more understanding about PHA biosynthesis and degradation. The production of PHA can be divided into two main steps: generation of hydroxy acyl-CoA and polymerization of hydroxy acyl-CoA into PHA [33]. In the first step, three main metabolic pathways allow the production of PHA, which are acetoacetyl-CoA synthesis, de novo fatty acid synthesis, and fatty acid *β*-oxidation [108]. Acetyl-CoA and acyl-CoA are common intermediates in all three pathways [35] and have an important role in the regulation of its production [31]. The general scheme of the three main pathways and some of the enzymes needed for the processes are represented in Figure 3.

Bacteria can produce PHA through sugar or fatty acid sources, varying the metabolic pathway used. When the main carbon source is sugar, bacteria can produce through two metabolic pathways: acetoacetyl-CoA synthesis (pathway I), whose reactions and enzymes required for the process will be discussed in the following section, and fatty acid biosynthesis (pathway II). During the reactions of the latter metabolic pathway, intermediates are formed that can be used as precursors in PHA synthesis, typically mcl-PHAs [121]. One of the compounds formed during the synthesis of fatty acids is (R)-3-hydroxyacyl-ACP, which can be converted to (R)-3-hydroxyacyl-CoA, via the acyl-ACP: CoA transacylase, encoded by the *phaG* gene [122]. The (R)-3-hydroxyacyl-CoA is an intermediate in the PHA formation process [121]. When PHA production originates from fatty acids as the main carbon source, the most common metabolic pathway is fatty acid β-oxidation. This pathway gives rise to several intermediates that can be converted into (R)-3-hydroxyacyl-CoA through the action of different enzymes, such as hydratases, epimerases, or reductases [121]. This allows an immense diversity in the production of mcl-PHAs. Using this pathway, it is also possible to produce scl-PHAs through the action of the enzyme enoyl-CoA hydratase (PhaJ) which converts enoyl-CoA intermediates to (R)-3-hydroxyacyl-CoA precursors [122]. All metabolic pathways end with the polymerization reaction through the enzyme PHA synthase (PhaC).

PHA metabolism of bacterial cells is controlled at multiple levels, via nutrient imbalances or stressors, and by specific regulators present in the *pha* gene cluster [123]. Different bacterial cell growth phases have been studied. Most bacteria that are PHA producers were shown to produce this polymer at the stationary growth phase. Some of them have been shown to form granules of crystalline PHB at those conditions [124]. To control the so-called PHA metabolic cycle, genome editing tools have been addressed to improve PHA production [125].

## 7. Bacterial Genes and Enzymes Involved in PHA Metabolism

In bacteria, PHA polymer is produced in the form of hydrophobic granules. These granules have several enzymes on their surfaces that are involved in polymer production, stabilization, mobilization, and degradation (Figure 4). Examples of some enzymes are phasins (PhaP, PhaI, PhaF, and others), which have a role in stabilizing the polymer, regulating proteins (PHA synthesis repressor-PhaR and PHB-responsive repressor-PhaQ) like the PHA synthase activating protein (PhaM) and an enzyme that promotes PHA degradation (PhaZ and PhaY) [33]. Phasins are not essential in the polymerization of PHA, however they play an important role in the regulation of the size, number, and surface-to-volume ratio of PHA granules [31].

Several enzymes participate in PHA production, PhaC being considered the key enzyme of this process [126]. This enzyme is present in all PHA-producing bacteria and there are four main classes, according to their substrate preference and their primary structure. Type I and type III PHA synthases are responsible for the production of scl-PHAs, while type II normally produces mcl-PHAs [109]. Type I and type II PHA synthases are characterized by an enzyme with only one subunit with 60–70 kDa and can be found in *R. eutropha* and *P. aeruginosa*, respectively [109,127]. Type III is present in *C. vinosum* and has two subunits (PhaC and PhaE), which together are responsible for polymer production [109,127]. Type IV is present in *Bacillus* species and comprises two subunits (PhaC and PhaR) that are essential for PHA production [33].

The two main enzymes responsible for PHA degradation are PHA hydrolase (PhaY) and PHA depolymerase (PhaZ). The degradation of PHA depends on a few factors such as the chemical composition of the polymer, polymeric chain length, crystallinity, and complexity [10]. Biodegradation of PHA can happen in aerobic or anaerobic conditions, resulting in different products. In the first case, PHA degradation results in carbon dioxide and water, and in the second, the products of PHA degradation are carbon dioxide and methane [4].

Oftentimes, the genes involved in PHA metabolism are inserted in gene clusters [35]. The most studied one is the *phaCBA* gene cluster which can be found in *R. eutropha* H16 [31]. The three genes present in this gene cluster (*phaA*, *phaB*, and *phaC*) encoded three proteins that are responsible for PHA production. This production is divided into three phases. Firstly, PhaA-*β*-ketothiolase (encoded by the *phaA* gene) catalyses the synthesis of acetoacetyl-CoA from acetyl-CoA. Then, PhaB-acetoacetyl-CoA reductase (encoded by the *phaB* gene) is responsible for the reduction of acetoacetyl-CoA to (R)-3-hydroxybutyryl-CoA, which is incorporated into the growing polymer by PhaC-PHA synthase (encoded by *phaC* gene) in the final step (Figure 3). Kutralam-Muniasamy et al. (2017) demonstrated the presence of other cluster groups in PHA producers. Examples of the various cluster groups present in PHA-producing bacteria are shown in Figure 5. Another gene cluster is present in many bacteria from the phylum *Firmicutes*, such as *B. cereus*. This cluster features the *phaC*, *phaB*, *phaR*, *phaQ*, *phaP*, and *phaJ* genes. The third cluster group is constituted by the genes *phaC*, *phaP*, *phaF*, *phaI*, *phaD*, and *phaZ*, having as a main example bacteria belonging to the genus *Pseudomonas* [128].

## 8. Strategies for Sustainable Production of PHA

PHA can be used in several applications, but it is in bioplastic manufacturing that these polymers have been most used. The current production of bioplastics represents only 1% of the total production of plastics [8]. The biopolymers market, especially the PHA market, needs to become competitive to start winning its space and replace the synthetic plastics market. To do this, several critical points in PHA production need to be addressed (Table 2). Currently, the cost of PHA production is between 2 and 5 times higher than the cost of synthetic plastic production [126]. The current situation of environmental pollution generated by plastics, aggravated by the COVID-19 pandemic, has increased the demand for bioplastics and its production is expected to increase in the next few years. In recent years, research in this area has focused on making the PHA production process a sustainable one. The research is mainly divided into upstream, fermentation, and downstream processes. In this review, the most commonly used upstream and downstream processes today will be discussed. The upstream processes usually handle the cultivation processes and feeding strategies, while the downstream processes consist of the polymer extraction processes of non-PHA biomass [112]. In general, researchers have been focusing on the improvement of upstream and downstream procedures, not only to increase the production of PHA, but also to increase the purity of the polymer obtained in a cost-effective way.

### 8.1. Upstream Processes in PHA Production

In the upstream procedures, research is essentially related to the discovery of new strains that produce PHA, or to the improvement of the production of PHA by new or existing strains already described through optimization of culture conditions and feeding strategies [33]. The fact that most current commercial PHB producers are heterotrophic [129] and that the production process is not economically sustainable leads to the search for other potential producers of the polymer. An example is the exploration of cyanobacteria as potential PHA producers. Some genera of cyanobacteria are already described as PHA producers, e.g., *Synechococcus*, *Nostoc*, *Spirulina*, etc. These bacteria have the main advantage that they do not require exogenous sugar for growth because they use CO_2_ for this purpose. The increase in CO_2_ in the last years has had negative impacts on the environment, namely the aggravation of global warming. Green technologies for the conversion and utilization of CO_2_ in a sustainable manner is a goal of today’s society in order to decrease its concentration and impacts on the environment; the use of cyanobacteria in PHA production is an example. Optimization of production parameters is also a useful tool for process sustainability and is the focus of several recent studies. In Chmelová et al. (2021), the effects of various culture medium components were studied through statistical design on the biomass production of *P. oleovorans*. This study allowed for the optimization of the production medium with minimal nutrients to increase PHA yielding [130]. Since 50% of PHA production cost is due to the cost of the carbon sources used in this process, some research focuses on investing in cheaper substrates including waste materials such as whey, starch, sugar-cane molasses waste, vegetable oils, and wastewater for the production of this polymer [131]. The use of waste in PHA production not only reduces production costs but also allows waste reduction, thereby reducing environmental pollution. In a recent study, it has been possible to verify a coupling between the two approaches mentioned above, the exploitation of cyanobacterial species for PHA production and the use of waste as a substrate for fermentation [132]. In that research, *N. muscorum* accumulated 31.3% CDW of P(HB-co-HV) from unrectified glycerol substrate derived from biodiesel industry waste.

Another approach to lower PHA production costs is the use of mixed microbial cultures (MMC) using WF as a substrate. The use of MMC has advantages such as cheaper carbon sources and no need for sterility control. However, when using MMC in PHA production, lower values of PHA content are obtained than those obtained by pure cultures [133]. In the last years, several studies have tried to optimize PHA production by MMC. The selection of bacterial strains and the culture conditions are extremely important for the success of this technique. Volatile fatty acids seem to be the best precursors for PHA production in MMC [120]. Shen et al. (2022) showed that the highest PHA productivity values to date were achieved using MMC with wastewater. The addition of pyruvate was the determining factor for the increased yield [134]. Recently, Vermeer et al. (2022) demonstrated the production of poly(3-hydroxyisobutyrate) (PHiB) using microbial enrichment cultures [135]. This study reveals the possibility of finding new metabolic pathways that generate new forms of polymers using MMC for PHA production.

Due to the great evolution in the field of molecular biology in the last decades, it is possible to manipulate bacteria at the genetic and metabolic levels to obtain desired products efficiently. Transcriptome and metabolome analysis allow us to understand the metabolic changes inherent in the production of PHA, thus facilitating better-targeted genetic manipulation to make the production process more efficient [33]. One of the most adopted ways to increase the production of the desired product is to hyper-activate the pathway that leads to the formation of this product and inhibit or eliminate competitive metabolic pathways. Some techniques, such as gene mutation, editing CRISP/Cas9 genes, and using strong promoters for essential genes for PHA production are used in an attempt to increase PHA production [35]. In Xiong et al. (2018), a genome editing method, the electroporation-based CRISPR-Cas9 technique, was developed to increase the efficiency of genetic manipulation of *R. eutropha*. Five putative restriction endonuclease genes were disrupted, and Cas9 expression was optimized enabling genome editing via homologous recombination based on CRISPR-Cas9 in *R. eutropha* [136]. Later, Jung et al. (2019) used the same technique in recombinant *E. coli* to improve PHA production. They deleted genes of competitive pathways and overexpressed a gene which catalyses the interconversion of NADH and NADPH, leading to a more efficient PHA production [137]. This technique may be used in the future by several producers to manipulate the carbon flux into PHA production.

Another widely used technique is genetic recombination which allows non-PHA-producing bacteria to produce the polymer [35], creating microbial cell factories. Typically, these bacteria are genetically well-studied, which facilitates genetic manipulation to increase PHA production. Recombinant organisms allow for solving some limitations of natural PHA producers such as long generation time, relatively low optimal growth temperature, and being hard to lyse. Another advantage is that recombinants usually lack the PHA degradation pathway, thus allowing for higher PHA yielding. One of the most commonly used non-PHA-producing bacteria in these types of studies is *E*. *coli*. This bacterium is considered a powerful microbial cell factory for PHA production on a commercial scale. By using different designs of recombinant *E. coli* strains, different types of PHA can be obtained. A novel PHA copolymer, P(3HB-co-3H2MB), was synthesized by constructing an artificial metabolic pathway in *E. coli* L55218. The copolymer was synthesized with various monomer compositions and their thermal properties were investigated. Last year, Goto et al. (2022) studied the effect of chaperones on PHA production in recombinant *E. coli* expressing PHA synthase from *B. cereus* YB-4. The results showed that chaperones have a positive action on the activation of PHA synthases and PHA productivity [135]. Recent studies also seek to find ways to optimize the conditions for PHA production by recombinant *E. coli* [39,138,139].

Pilot platforms for PHA production have been implemented in Treviso (northeast of Italy), Carbonera (northeast of Italy), and Lisbon to produce PHAs by open MMCs. These pilots exploited different organic biowastes as raw material for biodegradable polymers, characterized by thermal and chemical properties comparable to commercial plastics [140]. A pilot for production of PHA from paper industry wastewater was also investigated. The pilot plant was designed as a three-step process comprising anaerobic fermentation for maximization of the volatile fatty acid (VFA) concentration, enrichment of PHA-producing biomass, and accumulation for maximization of the PHA content of the biomass. They identified VFA production and pH control as major bottle necks [141].

### 8.2. Downstream Processes

Several processes are applied in PHA extraction and recovery from biomass after its production in bacteria. Problems related to downstream processes, such as high recovery costs and low PHA contents obtained, need to be overcome. The excessive use of environmentally harmful solvents is also a worrying factor that has been considered in current research. Usually, the extraction is carried out when the growth peak of the bacterial culture is reached to optimize the polymer yield [142]. The initial step usually consists of separating the biomass from the culture medium. This can be done by techniques such as centrifugation, filtration, and sedimentation, among others. The next step is cell disruption, extraction, and purification of the polymer. Depending on the technique chosen, the PHA is extracted by solubilizing it or by solubilizing the biomass together with the polymer. Then, the technique used for polymer purification will be dependent on the future application of the polymer. The solvent extraction method is one of the most widely used due to its simplicity and the high purity of the polymer obtained. This process has the advantage of eliminating impurities such as bacterial toxins [10]. In this process, the cells are pre-treated to facilitate the access and solubilisation of the polymer, usually in chloroform [143]. Then, the polymer is precipitated by the action of alcohols, such as methanol or ethanol. Despite being widely used at the laboratory level, its use is not recommended when high amounts of polymer are produced due to the vast number of solvents used and its toxic effects on humans and the environment [33]. Then, the need arose to find more eco-friendly ways to extract the polymer. In these methods, the cells can be digested through mechanical, enzymatic and/or chemical action and the PHA isolated through centrifugation or filtration [33]. Mechanical methods, such as high-pressure homogenization (HPH), are widely used in PHA extraction [144]. In this technique, cell lysis is based on the passage of biomass through a valve with a narrow orifice, followed by depressurization and a large increase in flow velocity, with consequent cavitation and high shear stress causing deformation/rupture of the cells in suspension [145]. The chemical digestion principle involves the solubilisation of non-PHA components using various surfactants, such as sodium dodecyl sulphate (SDS), Triton X-100, sodium hypochlorite, etc., after cell lysis [146]. In enzymatic digestion, enzymes such as lysozyme are used, allowing components such as proteins to dissolve. The PHA beads are separated by filtration or another method [146].

In large-scale production of PHA, halogenated extraction continues to be one of the most used methods that, in turn, results in higher production costs due to chemicals and the risk to operators and environmental contamination [147]. In addition to these methods, green solvents have been explored to substitute chlorinate solvents in PHA extraction, such as DMC. DMC is an acyclic alkyl carbonate that is currently produced in the industry [147]. Its versatility and non-toxicity are two properties that have increased interest in this solvent. In Samorì et al. (2015), *C. necator* freeze-dried biomass containing 74 ± 2 wt% of P(3HB) was extracted with DMC. This method was successful in the extraction of PHA from both freeze-dried biomass and highly concentrated microbial slurry without any pre-treatment before the extraction. The recovery and purity of the polymer were high, conserving the thermophysical characteristics of the polymers [147]. This reagent is more efficient in extracting and recovering PHA from MMC compared to more traditional methods [133].

Through this analysis, it is possible to see that there are several techniques currently used both upstream and downstream of the PHA production process. Their use will depend on the desired goals, and the right combination of the various techniques can lead to the desired PHA yield. Nowadays, researchers not only seek processes that result in higher PHA production but also processes that are more eco-friendly. Cost reduction, use of non-toxic reagents and solvents, and reuse of materials are important factors in developing industrially applied processes.

## 9. Current and Future Applications of PHA-Bacterial Based in Our Daily Life

Although PHAs are mainly used in the production of bioplastics, they can be used in many different areas, such as medicine, agriculture, industry, etc. The fact that many PHAs have properties like synthetic plastics, and the fact that they are biodegradable, eco-friendly, and non-toxic, are important factors in the implementation of these polymers in the plastics industry. Single-use plastics, namely used in packaging, are the main area of application of synthetic plastics. They are also proving to be a major source of environmental pollution. The properties of PHAs allow them to make materials with better oxygen, water, and fat/odor barriers than some conventional plastics such as PP [148]. PHAs should be a good substitute for conventional plastics in food packaging. Some copolymers of PHA are already approved by the Food and Drug Administration (FDA) and used in packaging materials and food additives, for instance [35]. The European Union has also recently updated the list of Food Contact Materials Regulation (EU) No 10/2011 on 11 January 2019, Commission Regulation (EU) 2019/37, which will boost the use of PHAs in packaging and food services. The copolymers PHBV and poly((R)-3-hydroxybutyrate-co-(R)-3-hydroxyhexanoate can be used on materials that come into contact with food. Currently, several companies are responsible for producing PHB and PHB composites for plastic manufacturing and packaging [32,149].

Another important characteristic of PHA that opens doors in the medical field, for example, is its biocompatibility. Currently, several studies are dedicated to studying the possibility of using PHA in various applications in the medical field. PHB-based nanocomposites have been explored in tissue engineering. In this area, porous materials that can mimic the physiognomy of bone have been explored, such as porous nanocomposites. In Mohan et al. (2021), PHB/OMMT nanocomposite showed advantageous characteristics when compared to neat PHB. Some of these improvements were in thermal stability and porosity, making this material a candidate for several applications such as 3D organ printing, lab-on-a-chip scaffold engineering, and bone tissue engineering [150]. PHB composites have also been explored in other applications, namely drug delivery and surgical applications [151,152,153,154]. Nanoparticles comprising liposomal particles enriched with PHB have been explored with the perspective to be used in cosmetic products [155].

In agriculture, PHA has mainly been used in the production of mulching material. The use of PHA-producing bacteria as plant growth promoters has also been reported to be beneficial. The fact that PHA is biodegradable is the main factor for its use in soils [127]. In the chemical sector, these polymers have been used in dye production, biodegradable solvents, and release agents for herbicides and pesticides [156]. Recently, the possibility of generating biofuels through the PHA synthesis pathway was demonstrated [35]. PHA can be a sustainable source for biofuel production due to the possibility of using several sources of waste as PHA precursors.

## 10. Conclusions and Future Perspectives

Several situations, COVID-19 being the most recent and evident, have highlighted the need for a viable solution to the use of synthetic plastics. Bioplastics have increasingly become a more sustainable solution, having as main their advantages biodegradability and biocompatibility. These properties enable their use in various applications, including packaging and medical applications. PHA is one of the most investigated biopolymers due to its similar properties to synthetic polymers, and its bacterial production has been highly exploited. The production of PHA at the industrial level has increased in recent years and is expected to continue to increase in the coming years. However, certain limitations such as the high cost of PHA production and other limitations in its production, recovery, and the physical and chemical properties of the final product may delay this evolution. Current research has tried to address these various limitations by presenting various perspectives and approaches. The search for new efficient producers, genetic engineering, and the use of MMC and cheap substrates for PHA production, have been the most frequent approaches to increase PHA yield and decrease production cost. Conventional techniques for extraction and purification of PHA from bacterial cells also pose sustainability problems due to the toxicity of the reagents involved. Recent research is seeking to replace these techniques with others that use green solvents or other less environmentally damaging approaches, seeking to maintain high percentages of recovery and purity of polymer. The increase in production and use of PHAs will have positive effects on the circular economy, promoting the valorisation of waste from various industries, thus making the process sustainable.

## Figures and Tables

**Figure 1 ijerph-20-02959-f001:**
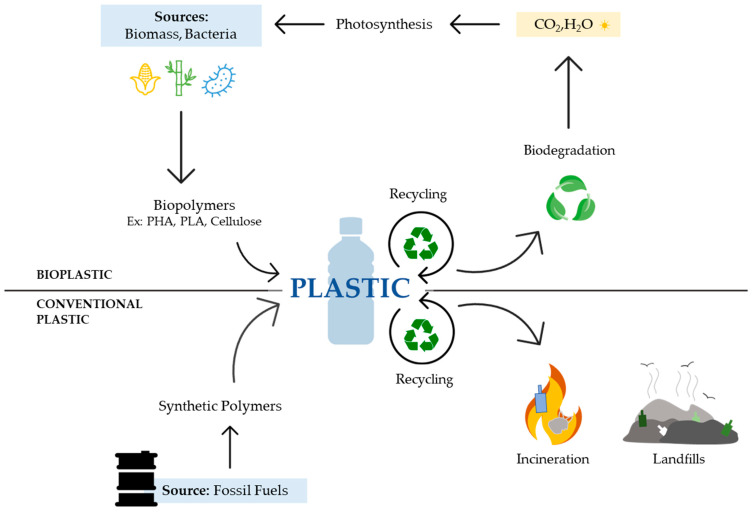
The classical plot of circulation of resources in the bio-based and biodegradable polymer life cycle.

**Figure 2 ijerph-20-02959-f002:**
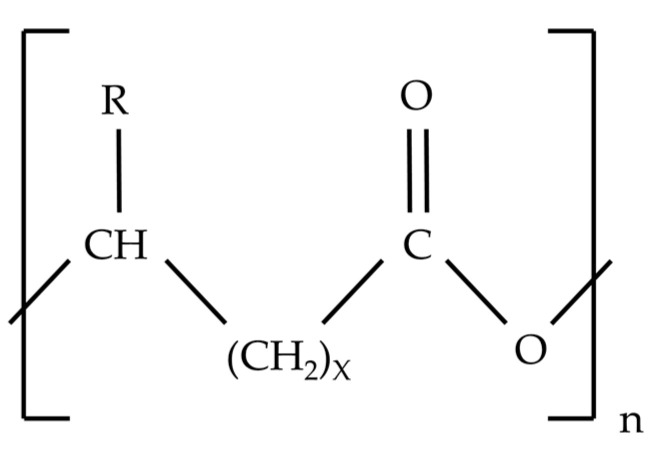
The general structure of PHA. X refers to the number of methylene groups in monomer composition and ranges from 1 to 8. *n* refers to repeating units of the polymer chain and ranges from 100 to 1000.

**Figure 3 ijerph-20-02959-f003:**
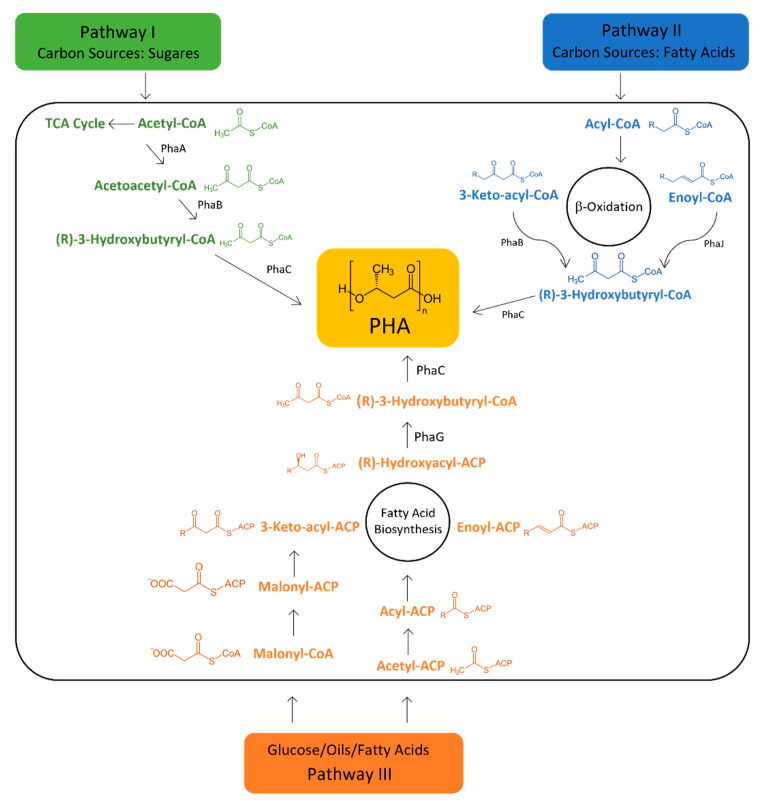
Metabolic pathways of PHA production. PHA can be produced by 3 main pathways: acetoacetyl-CoA synthesis (pathway I), fatty acid biosynthesis (pathway II), and fatty acid β-oxidation (pathway III). The metabolic pathway used depends on the bacterial strain and available substrates. All the metabolic pathways have the same final step catalysed by the essential enzyme—PHA synthase (PhaC).

**Figure 4 ijerph-20-02959-f004:**
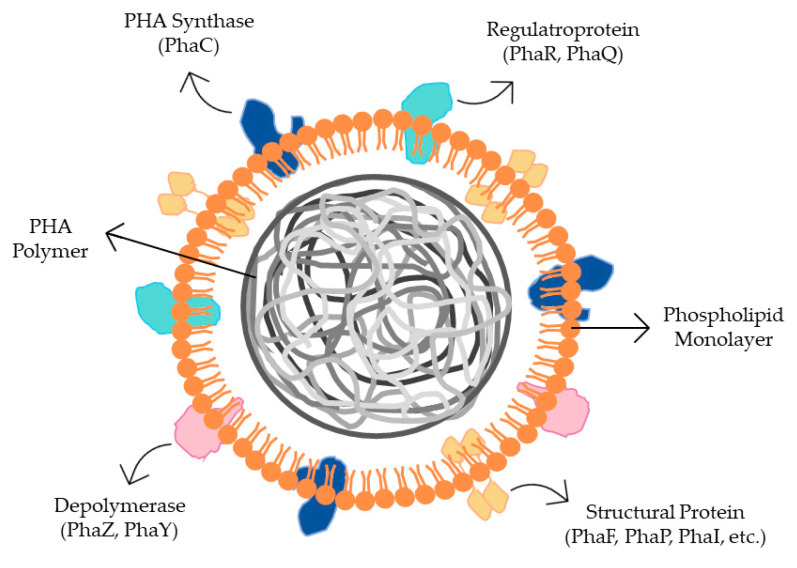
PHA granule with granule-associated proteins. The surface layer consists of a phospholipid monolayer that at its surface contains several proteins important in its production (PhaC), stability and mobility (PhaF, PhaP, PhaI, etc.), regulation (PhaR, PhaQ), and degradation (PhaZ, PhaY). This figure was generated by PowerPoint from Microsoft Office based on Muneer et al. (2020) [35].

**Figure 5 ijerph-20-02959-f005:**
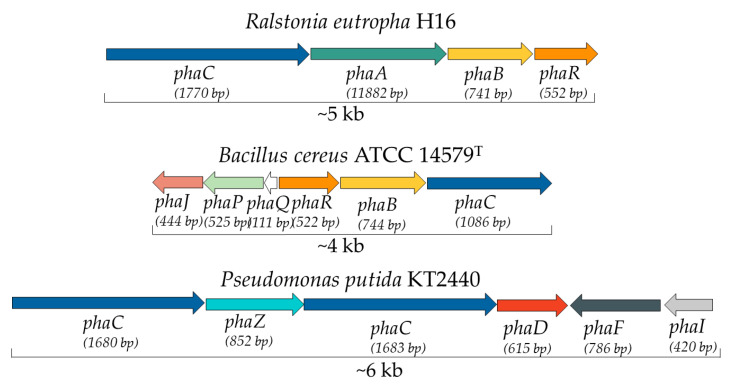
PHA gene clusters in different bacterial producers. *R. eutropha* H16 presents cluster *phaCAB*. *Bacillus cereus* ATCC14579^T^ presents genes *phaJ*, *phaP*, *phaQ*, *phaR,* and *phaB* clustered with *phaC*. Cluster *phaC1ZC2D* for mcl-PHA formation in *P. putida* KT2440 with *phaDFI* situated downstream of this cluster.

**Table 1 ijerph-20-02959-t001:** Examples of wild PHA-producing bacteria using various feedstock for PHA production.

Phylum (Class)	Genus	Substrate	PHA Type	PHA Yield (% dcw)	Reference
*Actinobacteria* *(Actinobacteria)*	*Micrococcus*	Glucose	PHB	56.59	[42]
	*Microlunatus*	Glucose	PHB	20–30	[43]
	*Nocardia*	Acetate, Succinate	PHB-*co*-HV	20	[44]
	*Rhodococcus*	Glucose	PHB-*co*-HV	40	[45]
	*Streptomyces*	Glucose	PHB	1.2–88	[46]
*Bacteroidetes* *(Sphingobacteriia)*	*Parapedobacter*	Molasses	PHB	50.24	[47]
*Cyanobacteria* *(Cyanophyceae)*	*Anabaena*	Sodium acetate	PHB	40	[48]
	*Aulosira*	Glucose	PHB	48.7	[49]
	*Chlorogloea*	Acetate	PHB	10.0	[50]
	*Nostoc*	Conditioning of effluent gases	PHB-*co*-HV	65	[51]
	*Spirulina*	Zarrouk medium	PHB	30.7	[52]
	*Synechococcus*	CO_2_	PHB	62	[53]
		Sucrose	PHB	17.15	[54]
*Deinococcus-Thermus* *(Deinococci)*	*Thermus*	Sodium gluconate, sodium octanoate	mcl-PHA	35–40	[55]
*Firmicutes* *(Bacilli)*	*Aneurinibacillus*	Glucose	PHB-*co*-HV	10–15	[56]
	*Bacillus*	Cheese whey	PHB	51.57	[57]
		Wheat starch wastewater	PHB-*co*-HV	59.5	[58]
		Lactate	PHB	64.7	[59]
		Xylose	PHB	62	[60]
	*Caryophanon*	Glucose	PHA	-	[61]
	*Geobacillus*	Glucose	PHB	68.9	[62]
	*Lysinibacillus*	Glucose	P(3HB-*co*-3HDD-*co*-3HTD)	-	[63]
	*Staphylococcus*	Hydrocarbons	PHB	15.2	[64]
*Firmicutes* *(Clostridia)*	*Clostridium*	-	PHA	26.75	[65]
*Proteobacteria* *(Alphaproacteobteria)*	*Bradyrhizobium*	YEM medium	PHB	13.95	[66]
	*Caulobacter*	Whey	PHB	31.5	[67]
	*Chelatococcus*	Glucose	PHB	44.5	[68]
	*Loktanela*	lignocellulosic biomass	PHB	78.3	[69]
	*Methylobacterium*	Methanol	PHB	52–56	[70]
	*Methylobacterium* *Methylocystis*	Methanol	PHB-*co*-HV	8.4	[71]
		Methane	PHB	51	[72]
	*Novosphingobium*	Glucose	PHB	80	[73]
	*Paracoccus*	*n*-hexanoic and *n*-octanoic	PHB-*co*-HV	49–61	[74]
	*Protomonas*	Methanol	PHB	64	[75]
	*Rhizobium*	L-Cysteine, L-Glycine, Arabinose, Glucose, Sucrose	PHB	5.5–70.0	[66]
	*Rhodobacter*	Acetate	Scl-PHA	50.8	[76]
	*Rhodospirillum*	VFAs	PHB-*co*-HV	-	[77]
	*Sphingobium*	Biomass hydrolysate	PHA	40.8	[78]
*Proteobacteria* *(Betaproteobacteria)*	*Alcaligenes*	Sucrose	PHB	88	[79]
	*Aquitalea*	Hydrocarbons	PHB	25–27	[80]
	*Azohydromonas*	Sucrose	PHB	95	[81]
	*Burkholderia*	Glycerol	PHB	81.9	[82]
		Sucrose + precursor GBL	P(3HB-*co*-4HB)	71.5	[83]
	*Delftia*	Acetic acid + precursor GBL	P(3HB-*co*-4HB)	52.0–60.0	[84]
	*Hydrogenophaga*	g-butyrolactone	P(3HB-*co*-4HB)	8.0	[85]
	*Hydrogenophaga* *Pandoraea*	Lactones	P(3HV-*co*-4HB)	18.0–26.1	[86]
		Kraft lignin	scl-PHA	60.0	[87]
	*Ralstonia*	Glucose	PHB	76.0	[88]
	*Ralstonia* *Schlegelella*	Glucose + Levulinic acid	PHB-*co*-HV	81.2	[89]
		Xylose + HV precursors	PHB-*co*-HV	35.5–68.9	[90]
*Proteobacteria* *(Deltaproteobacteria)*	*Desulfonema*	Benzoate	PHB	5.4–88.0	[91]
*Proteobacteria* *(Gammaproteobacteria)*	*Acinetobacter*	Rice mill effluent	PHB	94.3	[92]
	*Acinetobacter* *Aeromonas*	Rice mill effluent + VA	PHB-*co*-HV	85.9	
		Glucose + lauric acid	P(3HB-*co*-3HHx)	50	[93]
	*Aeromonas* *Azotobacter*	Lauric acid and valeric acid	P(3HB-*co*-3HV-3HHx)	71	[94]
		Coconut oil	PHB	49.6	[95]
		Glucose	PHB	85	[96]
	*Azotobacter*	Sucrose + valeric acid + sodium citrate	PHB-*co*-HV	68.1	[97]
	*Azotobacter*	Sucrose + PEG 300	PHB-PEG	34.2	[97]
	*Halomonas*	Glucose	PHB	81	[98]
		Galactose	PHB	78.1	[99]
		Sucrose	PHB-*co*-HV	80.1	[100]
	*Klebsiella*	-	PHA		[65]
	*Pseudomonas*	Hydrocarbons	mcl-PHA	25	[101]
	*Pseudomonas* *Saccharophagus*	Oils/Glycerol	mcl-PHA	8.0–61.8	[102]
		Cellulosic Waste	PHA	-	[103]
	*Salinivibrio*	Waste Fish Oil + Glycerol	PHB	51.7	[104]
	*Serratia*	Bicarbonate and glucose	PHV	-	[105]
	*Vibrio*	Acetate, glycerol, succinate, glucose and sucrose	PHB	1.0–45.5	[106]
	*Zobellella*	Glycerol	PHB	66.9–87.0	[107]

**Table 2 ijerph-20-02959-t002:** Future of microbial PHA production.

Critical Points	Opportunities	Benefits
Feedstocks	Alternative carbon sources	Cost reduction Waste management
Reduced diversity of producers	Assess microbial diversity (Biological Resource Centres)	New more productive pathways
Low microbial production	Microbial mixed cultures Microbial cell factories	Higher productivity
Bioindustry	New industrial production strategies Greener downstream processes	More efficient processes Reduction of environmental impact

## Data Availability

Not applicable.

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
