# Peer review of "The Role of Bacterial Polyhydroalkanoate (PHA) in a Sustainable Future: A Review on the Biological Diversity"

_ijerph, 2023, doi:10.3390/ijerph20042959_

Round 1
Reviewer 1 Report
It is an interesting review on PHA. However, there are some comments before its acceptance:
1- The language needs a professional revision. For example, Environmental problems should be Environmental challenges.
2- There are several good reviews on PHA. Therefore, it is important that in the introduction, the authors discuss the previous reviews and show the novelty of this work compared to them.
3- Section 2 is not much relevant to this work. I recommend moving it to section 1 while discussing the plastics in the society in general.
4- Some PHA-producing bacteria produce it along the cell growth, while some others produce it during the starvation phase. It should be clearly discussed and the metabolic pathways mentioned.
5- What about PHA production by mixed culture? You mentioned it, but is there any future in it?
6- It is still not clear what strains and what conditions should be used to reach affordable costs of PHAs. Any recommendation?
Reviewer 2 Report
Dear Authors, the review is very well organized and written. The used literature sources are very current that is the strongest advantage of prepared review. Moreover multidimensional analysis of current knowledge about PHA as possible replacement for synthetic polymers is also one from the main advantage of review. In my opinion the review can be published in its current form.
Reviewer 3 Report
This review article discusses about the role of bacterial polyhydroalkanoate (pha) in a sustainable future. Manuscript needs following changes to improve it further.
Comments:
1. Line 40: : recycling, incineration, landfill or abandoned…mention % of plastic managed through different methods.
2. In introduction: Need to add more information on negative effects of plastic and microplastics on environment, flora-fauna, and humans.
3. In introduction: Need to add more information on PHA and its copolymers, their application and various microbes and processes developed for their production.
4. Line 147: PHA are polyesters produced by many Gram-positive …what are main stress forces? Discuss in detail.
5. Table 1: Add a column mentioning PHA productivity also. Discuss and include other recent microbes reported for PHA production as suggested. Finding of novel polyhydroxybutyrate producer Loktanella sp. SM43 capable of balanced utilization of glucose and xylose from lignocellulosic biomass. Finding of novel lactate utilizing Bacillus sp. YHY22 and its evaluation for polyhydroxybutyrate (PHB) production. Finding of Novel Galactose Utilizing Halomonas sp. YK44 for Polyhydroxybutyrate (PHB) Production. Screening of the strictly xylose-utilizing Bacillus sp. SM01 for polyhydroxybutyrate and its co-culture with Cupriavidus necator NCIMB 11599 for enhanced production of PHB. Coproduction of exopolysaccharide and polyhydroxyalkanoates from Sphingobium yanoikuyae BBL01 using biochar pretreated plant biomass hydrolysate
6. Ralstonia is most widely studied for PHA production from various carbon sources. Authors has discussed only two examples. Discuss more recent work. Poly (3-hydroxybutyrate-co-3-hydroxyvalerate-co-3-hydroxyhexanoate) terpolymer production from volatile fatty acids using engineered Ralstonia eutropha. Poly (3-hydroxybutyrate-co-3-hydroxyhexanoate) production from engineered Ralstonia eutropha using synthetic and anaerobically digested food waste derived volatile fatty acids.
7. Line 196: Currently, several PHB copolymers….Biowaste-to-bioplastic (polyhydroxyalkanoates): Conversion technologies, strategies, challenges, and perspective
8. Line 202: Terpolymers have more than one secondary monomer in their constitution and are also currently 203 being explored as an alternative to copolymers…discuss recent reference…Tung oil-based production of high 3-hydroxyhexanoate-containing terpolymer poly (3-hydroxybutyrate-co-3-hydroxyvalerate-co-3-hydroxyhexanoate) using engineered Ralstonia eutropha.
9. Section 5: Information is too general. Need to add detailed information on various feedstocks, microorganisms used, PHA productivity and technoeconomic analysis and feasibility for large scale production.
10. In section 8.1: You should discuss case related to pilot and industrial scale.
11. Section 8.2: Discuss downstream methods used at large scale.
Round 2
Reviewer 1 Report
The revisions are acceptable.
Reviewer 3 Report
Authors have revised the manuscript according to the reviewers comments and recommended for publication as it is.